# A Multifunctional Polyethylene Glycol/Triethoxysilane-Modified Polyurethane Foam Dressing with High Absorbency and Antiadhesion Properties Promotes Diabetic Wound Healing

**DOI:** 10.3390/ijms241512506

**Published:** 2023-08-07

**Authors:** Chiu-Fang Chen, Szu-Hsien Chen, Rong-Fu Chen, Keng-Fan Liu, Yur-Ren Kuo, Chih-Kuang Wang, Tzer-Min Lee, Yan-Hsiung Wang

**Affiliations:** 1School of Dentistry, College of Dental Medicine, Kaohsiung Medical University, 100, Shih-Chuan 1st Road, Kaohsiung 80708, Taiwan; lori801107@gmail.com; 2Institute of Polymer Science and Engineering, College of Engineering, National Taiwan University, Taipei 106216, Taiwan; samsolvito@gmail.com; 3Division of Plastic & Reconstructive Surgery, Department of Surgery, Kaohsiung Medical University Hospital, Kaohsiung 80756, Taiwan; dr.chenrf@gmail.com (R.-F.C.); cell77821@gmail.com (K.-F.L.); kuoyrren@gmail.com (Y.-R.K.); 4Faculty of Medicine, College of Medicine, Kaohsiung Medical University, Kaohsiung 80708, Taiwan; 5Regenerative Medicine and Cell Therapy Research Center, Kaohsiung Medical University, Kaohsiung 80708, Taiwan; ckwang@kmu.edu.tw; 6Orthopaedic Research Center, College of Medicine, Kaohsiung Medical University, Kaohsiung 80708, Taiwan; 7Department of Biological Sciences, National Sun Yat-Sen University, Kaohsiung 80424, Taiwan; 8Academic Clinical Programme for Musculoskeletal Sciences, Duke-NUS Graduate Medical School, Singapore 169857, Singapore; 9Department of Medicinal and Applied Chemistry, College of Life Science, Kaohsiung Medical University, Kaohsiung 80708, Taiwan; 10PhD Program in Life Sciences, College of Life Science, Kaohsiung Medical University, Kaohsiung 80708, Taiwan; 11Graduate Institute of Medicine, College of Medicine, Kaohsiung Medical University, Kaohsiung 80708, Taiwan; 12Institute of Oral Medicine, National Cheng Kung University, No. 1, University Road, Tainan 701, Taiwan; 13School of Dentistry, National Cheng Kung University, Tainan 701, Taiwan; 14Taiwan Instrument Research Institute, National Applied Research Laboratories, Hsinchu 300092, Taiwan; 15Department of Medical Research, Kaohsiung Medical University Hospital, Kaohsiung 80756, Taiwan

**Keywords:** polyurethane, PEG, APTES, self-foaming reaction, porous structure, diabetic wound healing, negative pressure, high absorbency, antiadhesion

## Abstract

The delayed healing of chronic wounds, such as diabetic foot ulcers (DFUs), is a clinical problem. Few dressings can promote wound healing by satisfying the demands of chronic wound exudate management and tissue granulation. Therefore, the aim of this study was to prepare a high-absorption polyurethane (PU) foam dressing modified by polyethylene glycol (PEG) and triethoxysilane (APTES) to promote wound healing. PEG-modified (PUE) and PEG/APTES-modified (PUESi) dressings were prepared by self-foaming reactions. Gauze and PolyMem were used as controls. Next, Fourier transform-infrared spectroscopy, thermomechanical analyses, scanning electron microscopy and tensile strength, water absorption, anti-protein absorption, surface dryness and biocompatibility tests were performed for in vitro characterization. Wound healing effects were further investigated in nondiabetic (non-DM) and diabetes mellitus (DM) rat models. The PUE and PUESi groups exhibited better physicochemical properties than the gauze and PolyMem groups. Moreover, PUESi dressing showed better anti-adhesion properties and absorption capacity with deformation. Furthermore, the PUESi dressing shortened the inflammatory phase and enhanced collagen deposition in both the non-DM and DM animal models. To conclude, the PUESi dressing not only was fabricated with a simple and effective strategy but also enhanced wound healing via micronegative-pressure generation by its high absorption compacity with deformation.

## 1. Introduction

The management of wound exudate is an important part of wound management plans [1,2,3]. Especially in diabetic wounds, excess exudate is a common phenomenon that can cause maceration around the wound, delay healing and even cause other complications. The exudate contains water, electrolytes, nutrients, proteins, inflammatory mediators, matrix metalloproteinases (MMPs), growth factors, neutrophils, macrophages and platelets [4,5,6]. Generally, these components may assist healing and create a moist environment for the wound. The growth factors and proteases of the exudate complement control normal wound healing processes by forming new tissue and removing dead lesions to achieve wound repair [7]. However, several studies have reported that abnormal levels of growth factors and proteases were found in the exudate of diabetic foot ulcers and delayed wound healing. Excess wound exudate may increase the proteolytic activity of MMPs, which damages the wound bed, degrades the extracellular matrix and results in periwound skin problems [8]. In diabetic patients, chronic hyperglycemia adversely affects the immune system, neurological system and microvasculature, leading to various complications. Once a wound occurs, chronic inflammation perpetuates the inflammatory state and hinders the healing process. Persistent wounds in diabetic individuals are often attributed to chronic inflammation, which is primarily characterized by elevated levels of reactive oxygen species, proinflammatory chemokines and cellular pathogens. The impaired skin regeneration function further complicates the healing process in diabetic patients. Consequently, it is crucial to actively explore and identify innovative and effective approaches for treating diabetic wounds [9].

Additionally, wound exudate may act as culture medium for bacteria. Mustoe et al. demonstrated that an influx of polymorphonuclear leukocytes that release proteases and reactive oxygen species was caused by inflammation from the host immune response to bacteria [10]. Usually, an open wound becomes invaded by bacteria from the surroundings within 48 h, and biofilm overlying wounds protects bacteria from host defenses. When bacterial infection occurs, the infection may cause fasciitis, myositis and cellulitis [11,12].

A study found that continuous removal of exudate from wounds reduced bacterial counts and inflammation. The data also indicated that wound exudate was highly correlated with bacterial count and inflammation [10]. Therefore, wound dressings with high absorption activity may also reduce inflammation and bacterial infection.

Polyurethane (PU) foam is one of the most widely used wound dressing materials due to its excellent biocompatibility, water absorption capability, mechanical characteristics, physiochemical properties, gas permeability and unparalleled economic advantage [13,14,15,16]. Currently, there are many commercially available products that use PU material as the base material for dressings. However, different types of wounds may have various exudate rates and wound contamination risks. Challenges remain in terms of absorption efficacy for large fluid volumes in chronic wound dressing applications.

To enhance the absorption activity of PU foam, Iolanda Francolini et al. and Eugene Lih et al. conducted a series of studies that indicated that hydrophilic polymers, such as polyethylene glycol (PEG), prefer to absorb moisture upon contact with blood and other body fluids and expand to a volume that is several-fold larger than their original volume [17,18]. Additionally, Jeffery Lundin et al. synthesized PU hydrogel foams with different PEG molecules through a facile one-pot, solvent-free process [19]. However, these PEG modifiers were randomly combined with the soft and hard segments of the PU materials. Because the homogeneity cannot be effectively controlled, PU materials cannot maintain stable physicochemical activity and absorption capacity.

Moreover, unlike acute wounds, diabetic wounds or other chronic wounds suffer from chronic inflammation [20,21]. Therefore, reducing exudate reinfiltration and applying anti-tissue-adhesion wound dressings to avoid tissue damage during dressing changes are also important for promoting diabetic wound healing [22]. Several studies have noted that modifying the dressing surface by grafting aminoacyl groups, which has been used to prepare cellulose membranes or filter paper, may inhibit the adsorption of proteins, and bacteria have been correlated with both hydration and steric hindrance effects [23,24,25]. However, no studies have indicated that it is beneficial to modify materials with synthetic compounds as anti-adhesion wound dressings to promote wound healing.

In the present study, we aim to develop a multifunctional PU foam dressing that has high exudate absorption, does not allow the reinfiltration of exudate to wounds, and has antiadhesion and good tissue granulation properties to promote chronic wound healing. This is a new attempt at using PEG as the soft segment and APTES as the grafted aminoalkyl group within a two-step foaming reaction process (Figure 1).

We aimed to create a stable and homogeneous PU foaming process to produce a wound dressing with better physicochemical properties and exudate absorption efficiency. Furthermore, PEG-modified PU foam dressing (PUE) and PEG/APTES-modified PU foam dressing (PUESi) were synthesized with chemical cross-linking, forming network structures that had plentiful hydrophilic groups (Figure 2). We hypothesized that PEG modification would solve the low absorption problem of PU foam through the hydration of hydrophilic groups. Thus, PUESi would display better physicochemical properties and a higher absorbing capacity that would allow it to create a micronegative-pressure environment (Figure 1).

This study was conducted to evaluate the physicochemical properties of PolyMem (commercial PU foam product), PUE and PUESi in vitro. Additionally, the wound healing effects of Gauze, PolyMem, PUE and PUESi treatments were further investigated in nondiabetic control (non-DM) and diabetes mellitus (DM) wound healing.

## 2. Results

### 2.1. Synthesis and Preparation of PU Foams

The polyurethane foam was obtained in four stages, as shown in the reaction scheme in Figure 1. The first step consisted of the synthesis of the hydrophobic polyol, which had six hydroxyl groups, the reaction of TMP with HDI to obtain a triisocyanate intermediate, and the reaction of the triisocyanate intermediate with PPG 2000 (Figure 1a). Then, the hydrophilic diisocyanate was obtained by reacting HDI with PEG 2000 as a hydrophilic polyether diol (Figure 1b). Figure 1c shows that the hydrophobic polyol reacted with the hydrophilic diisocyanate to produce a prepolymer that included three to six isocyanate groups, and the prepolymer was modified with PEG 400 and APTES as an end-capping agent to obtain the PU foam sponge. In addition, the foaming reaction of the porous structure was induced by the reaction between the isocyanate (-NCO) and OH groups, which generated carbon dioxide (CO_2_) gas as a byproduct.

### 2.2. Chemical Synthesis Characterization and Morphological Observations

The FT-IR spectra of PolyMem, PUE and PUESi are shown in Figure 2a. Characteristic peaks of urethane bonding were detected for all samples, including peaks at 1600~1700 cm^−1^, 3400 cm^−1^ and 750~1150 cm^−1^, which were associated with C=O, N-H and C-O stretching vibrations, respectively. In addition, we also observed the disappearance of the -NCO band at 2265~2270 cm^−1^ in the spectra of the resulting samples, which was taken as evidence of complete polymerization.

The thermal stability and bound-water contents of the PU foams were determined by TGA, and the results are shown in Figure 2b. The traces of PolyMem pyrolysis were divided into three stages of thermal degradation. The first stage initiated at approximately 100 °C and was prolonged to 200 °C due to the loss of additives, such as carboxymethyl cellulose (CMC). The plateau between 200 °C and 320 °C was due to the loss of bound water from the soft segments. The third stage of pyrolysis occurred at approximately 320 °C to 420 °C due to the loss of hard segments. The traces of PUE pyrolysis were divided into two stages of thermal degradation. The first stage initiated at approximately 180 °C and was prolonged to 250 °C due to the loss of bound water from the soft segments. The plateau between 250 °C and 420 °C was due to the loss of hard segments. However, the Tg value of the PUESi group was approximately 500 °C, which was higher than that of the other groups because of the cross-linking and enhanced peak intensity variation in the urea linkages.

The morphology and microstructure of the PU foams were observed using SEM. As shown in Figure 2d, the cross-section images indicated that all the groups were characterized by a microporous structure, except PolyMem. Moreover, the PUE and PUESi foams had a three-layered, uniformly distributed porous structure, which included the oxidized top layer of PU, a middle layer with a pore size ranging from 100 μm to 500 μm and a bottom layer with a pore size ranging from 50 μm to 100 μm.

### 2.3. Mechanical Properties

An ideal wound dressing should possess excellent flexibility and mechanical strength so that it can protect wounds from physical damage and resist deformation caused by rubbing or collision. As shown in Table 1, the tensile strength and elongation of PUE and PUESi were significantly higher than those of PolyMem. However, there was no significant difference between those of PUE and PUESi.

### 2.4. Cytocompatibility

Biosafety is a crucial factor for a wound dressing; therefore, we meticulously investigated the cytotoxicity of the PU foams using a CCK8 test. Because keratinocytes and fibroblasts are the main components of cutaneous tissues, NIH 3T3 fibroblasts were chosen as the cell model for this assay. As shown in Figure 2c, NIH 3T3 fibroblasts that were treated with leach liquor that was extracted from PolyMem, PUE and PUESi foam showed cell viabilities (>100%) that were similar to that of the control group after 48 h, indicating that all groups exhibited no obvious cytotoxicity.

### 2.5. Water Uptake Ability

Sufficient water uptake capacity is essential for managing exudate, controlling infection and creating a moist wound environment that can increase the epithelialization rate. As shown in Figure 2e, the physiological saline uptake of PolyMem was 745% in 10 min, while the physiological saline uptake of PUE and PUESi reached 1435% and 1488% in 10 min, respectively, which were significantly higher than that of PolyMem. However, these results seem to be closely correlated with PEG modification or the 3D porous structures of the foams.

### 2.6. Moisture Vapor Transmission Rate (MVTR)

The MVTR is also an important indicator for whether a wound dressing can maintain an optimal moisture environment of the wound bed and reduce the possibility of wound infiltration. According to the wound dressing test methods (BS EN 13726-2), PUE and PUESi showed significantly higher transmission rates than PolyMem (Table 2). The data indicated that PUE and PUESi have stronger abilities to convert the exudate into moisture vapor after absorbing a large amount of exudate.

### 2.7. Swelling Characteristics and Micronegative-Pressure and Antiadhesion Properties

As shown in Figure 3a, we found that when the three wound dressings absorbed quantitative simulated body fluids (SBFs), only PUESi had significant absorption-dependent swelling, followed by PUE and PolyMem, with little difference between the swelling of the latter two wound dressings. Next, we investigated the absorption ability and swelling characteristics of the PU foams by conducting physics experiments and further investigated whether these PU foams could create a negative-pressure environment after absorbing exudate without fixation. After absorbing water from a centrifuge tube, it was found that only PUESi foams absorbed liquid quickly and changed shape to create a micronegative-pressure environment, as shown in Figure 3b.

There is a strong relationship between maceration and chronic ulcers. Therefore, we use the filter paper test to verify the surface dryness of the PU foams according to the CNS 12639-2020 test method. As shown in Figure 3c, the filter papers applied to PolyMem and PUESi were less wet, indicating good exudate management.

To determine the anti-adhesion property of the wound dressings, three groups of foam dressings were exposed to HRP-IgG, and the amount of absorbed protein is shown in Figure 3d. The PolyMem group adsorbed significantly higher amounts of HRP-IgG than the other two groups. Few HRP-IgG molecules were adsorbed by the PUE and PUESi groups. Additionally, there were no statistically significant differences between the two groups.

### 2.8. In Vivo Absorption and Anti-adhesion Tests

To study the effect of the anti-adhesion materials on wounds, we further performed studies with non-DM and DM wound models. The full-thickness wound regions were treated with PolyMem, PUE and PUESi foam dressings, using gauze as a negative control. The dressings were monitored, and their anti-adhesion properties were observed by taking pictures on day 3 postwounding. As shown in Figure 4, both non-DM and DM wound models provided similar results, except that the DM group had notable wound exudate, which resulted in a larger difference between each treatment group.

In the case of gauze treatment, gauzes are too hard for a new gauze dressing to be replaced easily, which may be due to the high vapor permeability of fiber yarns, tissue dehydration and gauze-tissue adherence. It was observed that exudates and blood oozed out from the waterproof membrane in the PolyMem treatment. This result implied that PolyMem may not provide sufficient absorption ability and may further cause wound maceration.

On the other hand, both the PUE and PUESi treatments exhibited clear and no tissue adhesion. These results indicated that the PUE and PUESi dressings with excellent absorption and anti-tissue-adhesion properties could be valuable wound dressing candidates.

### 2.9. Wound Healing Evaluation

The wound healing capability of the four types of wound dressings was further evaluated in animal wound models. All progressive changes in size in the remaining defect areas were observed and quantified, as shown in Figure 5. Optical images were taken, and quantification of the defects was performed for the non-DM and DM groups treated with PolyMem, PUE and PUESi foam dressings on days 0, 3, 7, 14 and 21 postwounding (Figure 5a,c).

In the non-DM group (Figure 5a), the wound region treated with PolyMem, PUE and PUESi foam dressings exhibited similar healing rates to those that were treated with gauze for the initial 7 days. However, in the case of PUE treatment, the wound regions showed significantly faster wound closure during a 14-day span than the other three groups. At day 21 postoperation, the remaining wound areas were significantly different among the different treatment groups. The PUE treatment showed a more significant decrease in wound area than the gauze and PolyMem treatments. However, there was no significant difference between the PUE and PUESi treatments.

In the DM group (Figure 5c), the gauze treatment showed a consistently slower recovery during a 7-day span, and it was apparent that abundant yellow purulent fluids permeated the STZ-induced wounds compared to those of the other three groups. However, on day 14 and day 21 postsurgery, the healing rates of the PolyMem, PUE and PUESi treatments were significantly higher than that of the gauze treatment. A post hoc comparison indicated a significant difference in the remaining defect areas between the test groups and the gauze group. As shown in Figure 5c, the wound regions in the PUESi treatment group were almost closed after 21 days, whereas the other three groups still exhibited considerable defect regions after the same period of treatment. These results indicated that the wound dressing effect on wound healing significantly differed between normal wounds and diabetic wounds, showing the importance of dressing selection for diabetic wounds.

### 2.10. Histological Analysis

To further evaluate the mechanism of wound healing and inflammatory reaction of the four types of wound dressings, skin tissues were collected and stained with H&E to assess the overall tissue features and CD45 of the inflammatory cells at day 7 postsurgery. As shown in Figure 6a,b, the wounds treated with PolyMem and PUE showed a significantly larger number of inflammatory cells in both the control and diabetes groups than those treated with PUESi and gauze.

Notably, the inflammatory infiltration in the gauze treatments was significantly lower or lower than that in the PolyMem treatments, both in the non-DM and DM groups. The result may be explained by the high-permeability fibers of the gauze that prevented maceration, thereby reducing the inflammatory reaction.

Collagen formation was observed by Masson trichrome staining. Wounded tissues were stained and quantified in the non-DM and DM groups treated with gauze, PolyMem, PUE and PUESi foam dressings at 7 days postwounding (Figure 7a,b and Table 3). Epithelized tissue was observed under different PU foam treatments, both in the DM and non-DM groups. Only in the gauze treatment group did the wounds show no keratin tissue in either the control or diabetes groups. This result indicated that the epidermal wound healing process had not yet occurred at 7 days postwounding.

Compared with the PUE and PUESi treatments, the PolyMem treatment showed less collagen deposition, not only in the non-DM group but also in the DM group. Especially in the DM group, the PolyMem treatment resulted in significantly less collagen deposition than the PUE and PUESi treatments. However, there were no significant differences in collagen deposition between the gauze and PolyMem treatments in either the non-DM or DM group. Additionally, PUESi treatment showed not only a significantly higher collagen density but also a more mature collagen morphology than the gauze, PolyMem and PUE treatments in the DM groups, according to the quantitative count results and staining images (Figure 7 and Table 3).

### 2.11. PUESi-Enhanced Epithelialization in Diabetic Rat Wounds

The epithelialization of wounds was recorded on the 21st day postwounding, as shown in Figure 8. In both the non-DM and DM groups, no epithelialization was observed in the wounds treated with gauze and PolyMem. Because of repeated damage from gauze, the regeneration tissue of the wounds in the gauze treatment group was flimsier than that of the other groups. Additionally, escharosis occurred in the wounds treated with PolyMem. However, in the non-DM groups, PUE and PUESi treatment resulted in epithelialization. Notably, there was a tendency for the formation of stratum basale in the PUESi treatment group, which indicated that the epithelialization caused by the PUESi treatment was more mature.

## 3. Discussion

Current therapies for difficult wounds, such as diabetic wounds, include various treatments and products. However, these wound care products or studies apparently suffer from numerous limitations, such as product fabrication difficulty, poor absorption ability, wound maceration and tissue-dressing adhesion, which may lead to a weaker wound bed, and secondary tissue damage can cause prolonged nonhealing of wounds and prolonged chronic inflammation [26,27,28,29]. PU foam is one of the most widely used biomedical materials because it can be produced with different characteristics according to its synthesis or production methods [30,31,32]. Recently, PU foam wound dressings have attracted much attention for application to difficult wounds because of their high economic advantage, absorption and biocompatibility. However, similar challenges remain in terms of the preparation of better functional PU foam wound dressings to increase their absorption ability and reduce tissue-dressing adhesion for clinical application.

A porous structure is important for enhancing the absorption ability of wound dressings. In previous studies, a porous structure was prepared by the combined methods of particle leaching, immersion precipitation and freeze-drying [33]. However, these preparation methods are inefficient and cannot produce a uniform porous structure. Thus, commercial products, such as Polymem, Mepilex and Convetec, are fabricated using different strategies to satisfy the demands of exudate management by combining additives or multilayer structures (Appendix A).

In this study, we developed a new synthesis procedure to prepare prepolymer and isoprepolymer first and then combined them with PEG400, H_2_O and APTES. The urethane groups were induced by the reaction between the -NCO and OH groups, the urea groups were generated by the reaction between the -NCO and amine (-NH2) groups, and the porous structure was created by the generation of CO_2_ gas as a byproduct of the reaction between the -NCO groups and water. The key to fabricating a sponge-like porous structure with multiple functions was the production of byproducts from the repeated reactions between the -NCO groups and water and the -NCO and amine (-NH_2_) groups. Such a design broadens the applications of the resulting foam and has the advantage of a manufacturing process (Appendix A). Thus, our new synthesis procedure for PUE and PUESi produced a unified porous structure by continuous self-foaming polymerization (Appendix A).

Exudate management of diabetic wounds is crucial to facilitate healing and achieve wound closure. However, exudate management is closely related to absorption ability and air permeability. Previous studies have indicated that the proteolytic activity of exudate is increased in chronic wounds. In particular, MMPs can not only degrade proteins to breakdown the extracellular matrix and damage the wound bed but also have inhibitory effects on growth factor activity [34]. Additionally, wound healing treatment with poor air permeability, which can provide a suitable maceration environment for bacterial infiltration and proliferation, may prolong the inflammatory response and delay wound healing [10]. Few studies or commercial products could solve both absorption and air permeability problems. For example, PolyMem probably applies carboxymethyl cellulose (CMC) as an additive to PU foam to increase absorption. This can be inferred from the gel-like discharge during the absorption test, product ingredients and TGA. ConvaTec, a multilayer composite dressing, combines PU foam and aquacel (hydrofiber) technology to provide an extra layer for absorption (Appendix A). However, the water uptake ability and MVTR results listed in Table 2 revealed that PolyMem had a lower absorption capability and air permeability than PUE and PUESi. This result indicated that PUE and PUESi can potentially provide a better wound healing environment than PolyMem.

In our study, the PEG-based PU structure and APTES chain absorbed water molecules when in a maceration environment (Figure 2). Additionally, due to the self-foaming reaction, PUE and PUESi had porous structures, as shown in Figure 2c, and they showed one-piece 3D porous structures, which included small pores on the wound-contacting surface and large pores in the absorbent layer. We speculate that the doubled absorption force exhibited by this 3D structure is related not only to modification by PEG and APTES hydration but also to the porous structure. Different from the commercial products, the products synthesized in this study show much better absorption ability (Figure 3a), air permeability (Table 2) and mechanical properties (Table 1) due to the complete polymerization of PU.

During wound treatment, filling a deep wound with gauze and then covering with dressing is a common clinical treatment [35]. The purpose is to overcome the limitation of the dressing that cannot touch the depth of the wounds so that the exudate can be better absorbed. According to the results of the absorption test shown in Figure 3b, we found that the PUESi group showed a striking effect of absorption with shape changing. This result may be explained by the linking of APTES chains after hydration. Thus, the pores of the foam structure that swell and diffuse outward probably fill the wounds when absorbing exudate (Figure 1). Clearly, this notion was proven in subsequent physical tests (Figure 3a) and animal experiments (Figure 4). However, for the other dressings, the swelling degree and deformation ability were not obvious after exudate absorption.

Tissue-wound dressing adhesion may induce pain and rewounding when changing the wound dressing. Some PU wound dressings have additives, such as glycerin or multilayer structures, to solve this adhesion problem. For example, Mepilex has a silicon sheet layer as an anti-adhesion layer. However, few studies have focused on the modification of PU material to reduce tissue-wound dressing adhesion. Previous studies have noted that hydrophilic polymers, such as PEG, tend to absorb moisture upon contact with blood and other body fluids. PEG-modified polymers resist the adsorption of proteins and bacteria, possibly via hydration and steric hindrance effects [17,36,37]. Thus, the modification of PU dressing with PEG and APTES was performed to produce PUE and PUESi. We speculated that the physical water barrier may generate a repulsive protein effect and further prevent tissue-wound dressing adhesion (Figure 2). Both the PUE and PUESi treatments reduced the problem of tissue dressing adhesion both in vitro (Figure 3d) and in vivo (Figure 4). Conversely, significant tissue-dressing adhesion was observed for the other wound dressing treatments, causing damage to the healing wound bed when changing the wound dressing.

Several studies have noted that vacuum can improve the clinical outcomes of wound closure [38,39,40]. One potential theory is that the application of suction to the wound not only evacuates interstitial fluid and cellular debris but also reduces local edema and decreases the likelihood of wound infection. Moreover, subatmospheric pressure increases blood flow and accelerates the formation of tissue granulation in the wound bed.

Interestingly, we found that when the wound dressing modified by PEG and APTES absorbed exudate, it simultaneously created a negative-pressure environment. As shown in Figure 3b, during the negative-pressure test, the wound dressing was attached to the centrifuge tube through negative pressure without any fixation. In other words, the rapid absorption and deformation of the wound dressing caused a pressure difference between the inside and outside the centrifuge tube, thereby generating a negative-pressure environment. Furthermore, we found that both non-DM and DM wounds showed lower degrees of inflammatory infiltration in the PUESi treatment group than in the other treatment groups at seven days, as shown in Figure 6. In addition, collagen deposition was denser and more mature than that in the other treatment groups, as shown in Figure 7. According to our in vitro and in vivo results, the wound healing processes of the PUESi treatment were consistent with those of negative-pressure treatment, as described in previous studies. Our results indicated that PUESi dressing may generate a micronegative-pressure environment on the wound region. Further studies will be required to investigate the mechanism by which PUESi dressing induces micronegative pressure. Although the wound healing effects of PUESi foam dressings were clearly observed in the animal models, the micronegative-pressure mechanisms are still not completely clear. Accordingly, the wound healing behaviors of PUESi require further evaluation with gene expression and angiogenesis in future investigations.

## 4. Materials and Methods

### 4.1. Materials

1,6-Disocyanatohexane (HDI) as a raw material for the hard segment, supplied by Aldrich Chemical Co. (USA), was dried at 60 °C in vacuum before use. Trimethylolpropane (TMP), polypropylene glycol 2000 (PPG 2000) and (3-aminopropyl)triethoxysilane (APTES), supplied by Sigma-Aldrich (Burlington, MA, USA), were dried at 60 °C under vacuum before use. Polyethylene glycol 400 and 2000 (PEG 400, 2000; E. Merck, India Ltd., Mumbai, India), supplied by E. Merck, India Ltd., as raw materials for the soft segment were dried at 60 °C in vacuum before use. PolyMem (a commercial polyurethane foam dressing used as a positive control sample) was purchased from Ferris Mfg. Corp. (Fort Worth, TX, USA). Gauze (as a negative control) and Tegaderm film (for dressing fixation) were purchased from 3M Company (Saint Paul, MN, USA).

### 4.2. Synthesis of PEG/APTES-Modified PU Foam Dressing

In step (a), TMP and HDI were mixed in a molar ratio of 1:3. The resultant mixture was stirred at 80 °C for 90 min under an enclosed nitrogen atmosphere. During the above reaction, Fourier transform-infrared (FT-IR) spectroscopy analysis was performed to monitor the presence of the characteristic NCO-group (-NCO) peak at 2270 cm^−1^. The characteristic NCO-group peak at 2270 cm^−1^ was successfully detected, indicating that a triisocyanate intermediate was obtained. Then, the triisocyanate intermediate and PPG triol (PPG 2000 triol) were mixed in a molar ratio of 1:3. The resultant mixture (prepolymer A) was stirred at 80 °C for 90 min under an enclosed nitrogen atmosphere. During the above reaction, FT-IR spectroscopy analysis was performed as described before. The characteristic NCO-group peak at 2270 cm^−1^ was not detected, indicating that a hydrophobic polyol with six hydroxyl groups was obtained.

In step (b), HDI and PEG diol (PEG 2000 diol) were mixed in a molar ratio of 2:1. The resultant mixture (isoprepolymer B) was stirred at 80 °C for 90 min under an enclosed nitrogen atmosphere. During the above reaction, FT-IR spectroscopy analysis was performed as described in step (a). The characteristic NCO-group peak at 2270 cm^−1^ was successfully detected, indicating that a hydrophilic diisocyanate was obtained. In step (c), the hydrophobic polyol and the hydrophilic diisocyanate were mixed in a molar ratio of 1:6, followed by stirring at 80 °C for 90 min under an enclosed nitrogen atmosphere. During the above reaction, FT-IR spectroscopy analysis was performed as described in step (a). The characteristic NCO-group peak at 2270 cm^−1^ was successfully detected, indicating that a prepolymer with 6 isocyanate groups was obtained. The prepolymer had a hydrophobic interior and a hydrophilic exterior. Then, the prepolymer, PEG 400, water and APTES were mixed in a molar ratio of 1:1:0.4:0.3:0.3. The resultant mixture was stirred at 20 °C for 10 s to 20 s under an enclosed nitrogen atmosphere to obtain a PEG/APTES-modified PU intermediate product. Then, the obtained PU intermediate product was coated on a PU film or PE release paper to obtain a PU foam sponge. Subsequently, the PU foam sponge was shaped and placed into sterilization bags. Sterilization was performed through gamma irradiation, conducted by CHINA BIOTECH CORPORATION, and gamma irradiation dose was 25 kGy at dose rate of 10 kGy/hr.

### 4.3. Characterization

The attenuated total reflectance-Fourier transform-infrared (ATR-FT-IR) spectra of the samples were obtained using an ATR-FT-IR spectrometer (SHIMADZU IR Spirit, Japan) [41]. The thermal degradation patterns of the PU foams were recorded with a thermogravimetric analyzer (TGA-55) under a N_2_ atmosphere from room temperature up to 650 °C with a heating rate of 20 °C/min. Analysis was carried out by gradually increasing the temperature and plotting the weight against the temperature. After the data were obtained, curve smoothing and other operations were performed to find the exact points of inflection. The morphologies of the PolyMem, PUE, PUESi foams were observed using scanning electron microscopy (SEM, JSM-6390LV, JEOL, Tokyo, Japan) [42].

### 4.4. Mechanical Property Measurements

The elongation and tensile strength (TS) of the PU foams were measured with a tensile test machine (506PC, Taiwan) according to the guidelines of ASTM standard method D 882-02. The samples (25 mm length × 100 mm width × 3 mm thickness) were fixed in a specific probe, and mechanical analysis was performed at a stretching speed of 10 mm/min with a preload of 0.5 N to determine the maximum load.

### 4.5. Water Absorption

The water uptake of the PU foams was determined by soaking them in phosphate-buffered saline (PBS, pH 7.4) at 37 °C. The weight of the specimens was measured after 2, 4, 6, 8 and 10 min. Water uptake was defined as the ratio of the increase in weight compared to the initial weight [35].
Water uptake rate (%) = [(W_1_ − W_0_)/W_0_] × 100
where W_0_ and W_1_ correspond to the weight (g) of the whole system at the initial time and after the time of analysis.

### 4.6. Micronegative Pressure

The micronegative pressure of PUESi was determined by an atmospheric pressure test. A 5 cm × 5 cm gauze, PolyMem, PUE, PUESi foam dressing and 15 mL centrifuge tubes with 2 mL red-stained physiological saline were prepared. Specimens were placed onto the top of a centrifuge tube and held in place while the tube was turned over for 10 s. Then, the hand of the experimenter holding the specimens in place was released to observe the negative pressure.

### 4.7. Moisture Vapor Transmission Rate (MVTR)

The moisture vapor transmission rate (MVTR) was measured according to the BS EN13726-2 guidelines. Clean, dry cylinders made of corrosion-resistant material with an internal diameter of 35.7 ± 0.1 mm (cross-sectional area of 10 cm^2^) with a flange at each end and capable of accommodating 20 mL of deionized water were prepared [43]. Specimens were cut into circles with a diameter of 55.6 mm and a thickness of 3 mm and placed in an incubator with a circulating fan under a constant temperature of 37 °C. The incubator had a design such that the air was distributed evenly throughout to maintain the relative humidity (RH) at less than 20% throughout the test. The MVTR was determined as follows:MVTR (g/M^2^·day) = (W_1_ − W_2_)/A × 24 h
where W_0_ and W_1_ correspond to the weight (g) of the whole system at the initial time and after the time of analysis. A corresponds to the transmission area of the sample (M^2^).

### 4.8. Surface Dryness

The surface dryness of the dressings was evaluated according to the guidelines of CNS 12639-2020. First, 3 mL double-distilled water (ddH_2_O) was dripped on specimens with dimensions of 30 mm length × 30 mm width × 3 mm thickness, and then the specimens were incubated for five minutes to ensure absorption. Six sheets of filter paper (ADVANTEC, Advantec Toyo Kaisha, Ltd., Tokyo, Japan) and 200 g weight was applied to the specimens. The surface dryness of the dressings was calculated according to the following equation:Surface dryness of dressing = W_d_ − W_i_
where W_i_ corresponds to the initial dry filter paper weight. W_d_ corresponds to the filter paper weight after the application of the filter paper to the specimens.

### 4.9. Protein Absorbance

Nonspecific protein binding to PUE, PUESi and the commercial wound dressing (PolyMem) was determined by horseradish peroxidase (HRP)-conjugated anti-immunoglobulin G (IgG) adsorption quantification. All experiments were performed in triplicate. Specifically, the foams were cut into disks with a diameter of 7 mm and height of 2 mm, and then they were soaked in 125 μL of HRP-IgG solution (2 μg/mL) at 37 °C for 90 min followed by five rinses with PBS to remove the nonadhered proteins. The foams were then removed and placed in 24-well plates, and 250 μL 3,3′,5,5′-tetramethylbenzidine (TMB) substrate (Thermo Scientific™, Thermo Fisher Scientific Inc., Waltham, MA, USA) was then added. Enzyme activity was stopped by adding an equal volume of 2 M H_2_SO_4_ after 5 min. The tangerine-colored solution (i.e., the relative protein adsorption) was measured at 450 nm [22,33].

### 4.10. In Vitro Evaluation of Cytotoxicity

The cytotoxicity of the PU foams against NIH 3T3 fibroblasts was investigated according to ISO10993-5 [44]. First, the leach liquor was obtained as follows: sterilized specimens (1.4 cm diameter) were immersed in 1 mL of Dulbecco’s modified Eagle’s medium (DMEM; Invitrogen, Carlsbad, CA, USA) at 37 °C for 24 h. Next, NIH 3T3 fibroblasts were inoculated into each well of a 96-well plate and incubated for 24 h at 37 °C. The medium was then replaced with 500 μL of leach liquor or pristine medium containing 10% fetal bovine serum (FBS; HyClone, South Logan, UT, USA). Cells inoculated with pristine medium were used as a control group. After 24 and 48 h, the medium was replaced with 100 μL of pristine medium containing 10 μL of CCK8 solution (Dojindo Laboratories, Kumamoto, Japan) and then incubated at 37 °C for 2 h. Finally, the optical density at 450 nm was measured using an enzyme-linked immunosorbent assay reader, and the cell viability was calculated.

### 4.11. In Vivo Animal Tests

The in vivo study was performed under an approved protocol (no. 12-0226) of the Institutional Animal Care and Use Committee (IACUC) of Kaohsiung Medical University. The use and handling of the animals was performed according to the Guide for the Care and Use of Laboratory Animals of Kaohsiung Medical University.

### 4.12. Streptozotocin (STZ)-Induced Diabetes

Dorsal full-thickness skin wound defects were created in a streptozotocin (STZ)-induced diabetic rodent model [45,46,47,48]. Diabetes was induced in 8-week-old Wistar rats (BioLASCO Taiwan Co. Ltd., Taipei, Taiwan) by a single intraperitoneal injection of STZ (50 mg/kg; Sigma-Aldrich, St. Louis, MO, USA). All animals were confirmed to be diabetic (plasma glucose levels of >300 mg/dL) at 1 week after STZ injection.

### 4.13. Experimental Design

The in vivo experiments with PU foam dressings included 20 nondiabetic (*n* = 5) and 16 diabetic (n = 4) 8-week-old male Wistar rats and were performed to validate the improved wound healing capabilities of PUESi and to compare them with those of gauze, PolyMem and PUE. All animals were pathogen free and were maintained using the same dietary and environmental conditions. After a weeklong adaptation period, healthy animals were selected for use in the experiments. After administering 5% isoflurane (AbbVie Inc, North Chicago, IL, USA) anesthesia and carrying out hair removal, two full-thickness square wounds with dimensions of 20 mm × 20 mm and a depth ranging about 1~1.5 mm were created on the dorsum of each animal, and the corner for the wound defect was sutured in place with 4-0 silk sutures [49,50,51,52,53,54]. One wound was designated for in vivo sampling, while the other wound was utilized for photographic documentation to observe wound healing progression.

Each wound was covered with one of the four dressing materials, gauze, PolyMem, PUE and PUESi, with square dimensions of 30 mm × 30 mm. For the fixation of the dressing materials, each wound was temporarily covered with transparent Tegaderm^TM^ (3M HealthCare, Borken, Germany) and subsequently fixed with Coban^TM^ (3M HealthCare, Borken, Germany).

### 4.14. In Vivo Anti-Adhesion Test

Optical images of all wounds were taken to assess tissue adhesion 3 days after surgery. The anti-adhesion results were determined according to the force at which the dressing was removed, the tissue adhesion and whether the tissue was bleeding.

### 4.15. Wound Size Assessment

Optical images of all wounds were taken to assess wound healing at 0, 3, 7, 14 and 21 days after surgery [55,56]. The size of the wound area was quantified using the National Institutes of Health ImageJ 1.36b imaging software (National Institutes of Health, Bethesda, MD, USA). To compare the defect area to the original wound size at day 0, the ratio of remaining wound area was calculated using ImageJ software for the area measurements.

### 4.16. Histological Analysis and Immunohistochemistry

Full-thickness 3-mm biopsies were performed from the wound margin on days 7, 14 and 21 [33]. Rats were terminated 21 days postsurgery. The tissues from the wound area, including the surrounding normal skin tissue, were harvested in bloc. The tissue specimens were fixed in 10% formalin (Sigma-Aldrich) and embedded in paraffin. Sections (4 μm thick) for each group were stained according to three protocols: hematoxylin and eosin (H&E), Masson’s trichrome (MT) and CD46 [35,57].

### 4.17. Statistical Analysis

The experimental data were analyzed using one-way ANOVA with Tukey’s post hoc test for multiple comparisons. * and **, represent *p* values < 0.05 and 0.01, respectively, all of which were considered statistically significant.

## 5. Conclusions

Our study provides a simple two-step procedure to produce a uniform PU foam dressing. Our results also show that the PEG/APTES-modified PU wound dressing PUESi provides multiple superior functions of high absorption and anti-tissue-adhesion in vitro. Interestingly, following absorption, the PUESi foam structure changes and swells outward, which allows the wound structure to be fitted and further generates a micronegative pressure for wound healing. Furthermore, the PUESi dressing revealed better wound healing properties, less inflammatory infiltration and enhanced collagen deposition in both non-DM and DM animal models.

## Data Availability

All data collected or analyzed during this study are included in this article and its Appendix A.

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
