# Peer review of "A Multifunctional Polyethylene Glycol/Triethoxysilane-Modified Polyurethane Foam Dressing with High Absorbency and Antiadhesion Properties Promotes Diabetic Wound Healing"

_ijms, 2023, doi:10.3390/ijms241512506_

Round 1

Reviewer 1 Report

1. Experiments that may need to be supplemented include: Evaluating the effects of dressings on cell adhesion, proliferation, and migration to understand their cellular-level mechanisms for wound healing. And some of the pictures in the paper are blurred. This may affect the reader's accurate understanding of the experimental results. You can edit the picture again to ensure clarity.

2. The practical application is, how do I store the foam dressing? What is the difference between a dry dressing and a wet dressing?

3. The Figures in the article should be arranged more closely, preferably without blank sections to save space (such as Figure 5).

4. Background descriptions for wound healing can be strengthened by citing 10.1016/j.cej.2023.141852; and what are the advantages of the current work compared to published articles?

5. There are some formatting errors in the article. For example, spelling of references must be checked to meet the journal style (such as Reference 12). Please check carefully and use abbreviation properly.

6. Does PEG/APTES-modified PU match the wound healing process?

ns

Reviewer 2 Report

The article presents the synthesis of modified polyurethane in applications such as wound dressings. The material is novel and in vitro and in vivo results reveal that it is an appropriate material for this purpose.

Reviewer 3 Report

In this study, the authors made new dressing materials (PUE and PUESi) and showed superiority of these materials with many experiments. Most of the experiments were well performed and the results support efficacy of the authors’ new materials. However, there are some concerns in the manuscript. My concerns are listed below.

1: The major concern is lack of evidence for anti-adhesion and micro-negative pressure. Because there are no quantitative data in vivo, involvement of anti-adhesion and micro-negative pressure in the efficacy of the new materials is just speculation. The properties of anti-adhesion and micro-negative pressure should be omitted from the title and the abstract. Such properties should be mentioned only in Discussion.

2: Abbreviations (PEG/APTES) should be avoided in the title.

3: In the abbreviations of APTES, PUE, and PUESi, what does each character stand for? More details should be shown when the words first appear in the main tex.

4: In the legend of Fig. 2, the name of (e) is lacking.

5: In the result section of 3.10, the DM group is shown in Fig 5c and 5d. ‘In the DM group (Fig. 5b)’ should be changed to ‘In the DM group (Fig. 5c)’. ‘As shown in Fig. 5b’ should be changed to ‘As shown in Fig. 5c’. 

Reviewer 4 Report

Great introduction to setup the article, with good use of schematic diagram

Is water absorption ability relatable to wound exudate absorption due to viscosity differences?

You used PBS for the water absorption and ddH20 for surface dryness, is it better to use the same solution eg both water or PBS?

Some information about the STZ model in the introduction could be beneficial

Section 2.13 – I’m guessing there were 4 wounds on each mouse to get the 20 from n=5? This section needs rewriting slightly to be clearer about the conditions, 4 treatment conditions – with 5 repeats for healthy and 4 repeats for diabetic.

Section 2.14 – how do you record the force required to remove dressing with optical image? What equipment do you use?

Section 3.4 – you state results after 72 h but the graph shows 24 h + 48 h?

Beneficial results with high potential. 

Some grammatical errors that need correcting. 

Round 2

Reviewer 3 Report

The manuscript improved after revision.

Author Response

Thank you for acknowledging the improvement in the manuscript after revision.